# A Spectrum of Solutions: Unveiling Non-Pharmacological Approaches to Manage Autism Spectrum Disorder

**DOI:** 10.3390/medicina59091584

**Published:** 2023-08-31

**Authors:** Arunima Mondal, Rashi Sharma, Umme Abiha, Faizan Ahmad, Anik Karan, Richard L. Jayaraj, Vaishnavi Sundar

**Affiliations:** 1Department of Human Genetics and Molecular Medicine, Central University of Punjab, Ghudda 151401, India; 2Department of Biotechnology, Delhi Technological University, Bawana, Delhi 110042, India; 3IDRP, Indian Institute of Technology, Jodhpur 342030, India; 4All India Institute of Medical Sciences, Jodhpur 342005, India; 5Department of Medical Elementology and Toxicology, Jamia Hamdard University, Delhi 110062, India; 6CL Lab LLC, Gaithersburg, MD 20878, USA; 7Department of Pediatrics, College of Medicine and Health Sciences, United Arab Emirates University, Al Ain 15551, United Arab Emirates; 8Department of Internal Medicine, University of Nebraska Medical Center, Omaha, NE 68198, USA

**Keywords:** neurodevelopmental disorder, ASD, neurotherapeutics, nutritional therapy, microbiome

## Abstract

Autism spectrum disorder (ASD) is a developmental disorder that causes difficulty while socializing and communicating and the performance of stereotyped behavior. ASD is thought to have a variety of causes when accompanied by genetic disorders and environmental variables together, resulting in abnormalities in the brain. A steep rise in ASD has been seen regardless of the numerous behavioral and pharmaceutical therapeutic techniques. Therefore, using complementary and alternative therapies to treat autism could be very significant. Thus, this review is completely focused on non-pharmacological therapeutic interventions which include different diets, supplements, antioxidants, hormones, vitamins and minerals to manage ASD. Additionally, we also focus on complementary and alternative medicine (CAM) therapies, herbal remedies, camel milk and cannabiodiol. Additionally, we concentrate on how palatable phytonutrients provide a fresh glimmer of hope in this situation. Moreover, in addition to phytochemicals/nutraceuticals, it also focuses on various microbiomes, i.e., gut, oral, and vaginal. Therefore, the current comprehensive review opens a new avenue for managing autistic patients through non-pharmacological intervention.

## 1. Introduction

In 1908, a Swiss psychiatrist named Eugen Bleuler invented the terminology of “autism”, which originated from the ancient term “autós”, which signifies “self”, to characterize the detachment from reality of patients with schizophrenia [1,2,3]. Leo Kanner used the phrase in 1943 to describe linguistic and social isolation problems in children who did not have psychosis or other psychological illnesses. Such children struggled to engage and communicate with others, had a specific pattern of behavior, and were uninterested in social affairs [4,5,6,7,8]. One out of every 88 children has developmental difficulties, and this percentage seems to be rising. The frequency of autism in males and females is equivalent to approximately 5:1, affecting about 1.5% of the population [9,10,11,12,13,14,15]. The pathophysiology of ASD is not entirely known, and comorbidities, including epilepsy, attention, mood, and language impairments; sleep disturbances; gastrointestinal issues; and intellectual disability, are frequent (70% of cases). ASD is believed to be a developmental defect of brain processes brought on by genetic and neurological reasons, creating social disruption, which results in limited attentiveness and compulsive behaviors [16,17,18,19,20]. An aberrant gene gets “turned on” during the early stages of fetal development, altering the body. Its expression can be changed without modifying the primary DNA sequence of other genes. The pathogenesis of ASD, which appears to be primarily driven by heterogeneous genetic mutations and variants and modulated by diverse gene–environment interactions, including pregnancy-related factors (such as maternal immune activation, maternal toxins, and perinatal trauma), may be a significant factor in the absence of disease-modifying therapies. Currently, there are few accessible pharmacological and non-pharmacological methods for ASD intervention. Different psychiatric drugs are used as pharmacological interventions, whereas specialized foods, herbal supplements, chiropractic adjustments, art therapy, mindfulness practices, and relaxation techniques are a part of non-pharmacological methods. As there are not any particular behaviors that aid in identifying people with ASD, it does not have a single management strategy. In addition to this, the cost of management of an autistic individual for a lifetime, as estimated in a study conducted in the USA, approximately amounts to around USD 3.6 million, which goes up as the case worsens. Apart from this huge cost, the constant care and support required are beyond estimation and are not a treatment that everyone can afford. Many parents have always turned to alternative therapies to help autistic children [21,22,23,24,25,26,27,28,29,30,31,32,33,34,35,36,37,38]. For different patients of ASD, specific CAM therapies, which include essential fatty acids, vitamins, an oligoantigenic diet, herbal remedies, and amino acids, are found to give favorable results. ASD nutritional dysfunctions should be considered part of the therapy/management process, as managing autism care is a complex condition for individuals and their families [39,40,41,42]. In this review, we focused on non-pharmacological interventions like different diets, supplements, camel milk, hormones, etc., and we also included different microbiomes, i.e., oral, vaginal, and gut microbiomes. This review will reveal a new horizon for the treatment and management of ASD. All non-pharmacological interventions are easily available on the market and are affordable, too. All non-pharmacological interventions can be used in combination with a low dose of pharmacological interventions, i.e., aripiprazole and risperidone. Non-pharmacological treatment has a far better response than relying on only drugs, as these drugs show adverse effects like weight gain, blurred vision, low blood pressure, seizures, low white blood cell count, drowsiness, dizziness, restlessness, dry mouth, constipation, and nausea. All non-pharmacological interventions mentioned in this review will help to manage the symptoms of ASD without showing adverse effects and are discussed in the coming sections of this review.

## 2. Diets for ASD

There is a need for additional management options that can improve outcomes for individuals with ASD. Different dietary supplements for the management of ASD are discussed below.

### 2.1. Elimination Diet for ASD

As the term signifies, some foods are avoided in the diet on the theory that particular ASD symptoms are related to foods that appear to be impacted by dietary hypersensitivities [43]. Such foods create gastrointestinal issues (GI) issues and raise IgG levels as the individual may be sensitive to the foods or their additives [44]. IgE and IgA antibody types have already been linked to immune dysfunction in people with autism. Diets must be closely controlled because removing foods to which an individual is allergic can cause malnutrition, which can worsen the symptoms of the disease by causing anemia. Findings show that adopting an exclusion diet regimen considerably improved the pathogenic alterations in autistic patients [45]. The popular elimination diet (gluten-free casein-free diet, GFCF) removed the proteins included in milk and cereals. The aforementioned diet calls for a decrease in or total removal of all the above-listed proteins [44]. Cows’ milk, cheese, and other dairy products contain casein, which, when removed, can cause a calcium deficiency since it is a crucial nutrient for bone and tooth health. Alternatives such as goat or sheep milk are frequently recommended but might require the body to confront new allergens [45]. The elimination diet can lead to malnutrition if not carefully monitored, while the specific carbohydrate diet may be challenging to follow and restricts certain foods that are important for overall health. Additionally, some nutritional supplements may interact with medications or have harmful side effects if taken in excessive amounts. 

### 2.2. Casein and Gluten for ASD

Gluten, milk, barley, rye, and wheat include casein, which has anti-inflammatory characteristics that regulate immune responses [46,47]. In particular, in persons with ASD, casein and gluten can promote the production of antibodies against IgA and IgG, worsening their immune dysregulation. The small intestinal mucosa works as a luminal barrier, keeping germs out, and such compounds are not permitted to enter the circulatory system. People with ASD, on the other hand, have higher intestinal permeability to such compounds, resulting in inflammation [48,49,50,51,52]. Casein and gluten products need to be consumed based on a clinician’s advice, as these diets can cause inflammation at high doses, which can lead to other disorders. 

### 2.3. Specific Carbohydrate Diet for ASD

A study by Gottschall, E. (2004) popularized this diet as a method of autism management. This diet’s central premise is to prevent the advancement of pathogenic intestinal microflora’s by alleviating malabsorption [53,54]. This diet recommends consuming monosaccharides, like those found in fruits, vegetables, and honey, rather than complex polysaccharides because polysaccharides take longer to digest [53]. Difficult polysaccharide digestion disrupts gastrointestinal tract function, resulting in absorption difficulty and the accumulation of left-over food. Intestinal pathogenic flora thrives in this food-accumulated environment [54]. This diet aims to help individuals lose weight, restore normal intestine functions, and minimize intestinal cancer formation. Meat, eggs, natural cheese, vegetables (pepper), cauliflower, onions, cabbage, spinach, homemade yogurt, fruits, nuts (walnuts, almonds), beans, and soaked lentils are all excellent protein sources and are recommended. Complex carbohydrates (e.g., sugar) are prohibited in the specific carbohydrate diet [54,55]. Only those foods that require minimal digestion are allowed. In a study conducted by Żarnowska et al. (2018), this diet was followed by people with Crohn’s disease, both colonic and ileocolonic [55]. Symptoms improved after three years of monitoring. These findings may be generalizable across populations of people with ASD. Learning and memory were also highly improved, as were responsive and imaginative language difficulties [55]. Carbohydrate diets need to be consumed based on a clinician’s advice and at the recommended dose as complex carbohydrate diets can cause different complications like inflammatory bowel disease (IBD). 

### 2.4. Ketogenic Diet for ASD

The ketogenic diet is a general term for a low-carbohydrate, moderate-protein, and high-fat diet that encourages our body to use ketones instead of glucose for energy. This results in more ketones in the blood, reduced blood glucose, and better functioning of mitochondria [56,57]. This diet has shown potential in treating patients with refractory epilepsy, which is considerably more typical in persons with ASD than those without ASD and other related nervous system problems [58,59]. A study by Kasprowska-Liśkiewicz D. et al. (2017) revealed that their sample exhibited fewer seizures and superior learning and social abilities [60]. El-Rashidi, O. et al. (2017) studies in people with ASD also revealed that the medication produced moderate improvements [61]. A ketogenic diet produces better responses and fewer complications in comparison to the elimination diet, casein and gluten diet, and carbohydrate diet, but still, the dose needs to be set by a clinician/dietician to avoid complications. 

## 3. Nutritional Supplements for ASD

Numerous studies have suggested that poor behavioral evaluation test scores are continuously connected to low nutritional fulfillment. Hyperactivity, agitation, and irritability decrease when certain nutrient supplements are administered. Impulsivity and the inability to pay attention both improve dramatically [62]. During the day, a diverse mix of vegetarian and animal proteins is consumed to meet the daily need for amino acids. Amino acids (AAs), which have long been the basic building blocks of our bodies, make up proteins. The body may synthesize certain amino acids, but amino acids should be acquired from protein-rich diets [62]. The effect of nutritional therapy on ASD is shown in Figure 1. According to a study, neuroactive amino acids play a vital role in central brain activities. Neuroactive AAs are crucial in etiology and play a part in treating autistic symptoms [63]. It is also essential to watch for changes in their bodily fluid concentrations and see whether they correspond to early signs. Their availability, metabolism, and receptor functionality must all be considered [63]. They have been connected to the causes and therapies of numerous mental illnesses. More research is needed to see if other amino acids are involved. Discussed below are a few nutritional supplements.

### 3.1. Omega-3 Fatty Acids for ASD

Omega-3 fatty acids are polyunsaturated fatty acids (PUFAs) recognized as -3 fatty acids or n-3 fatty acids. Triglycerides and phospholipids are two natural forms of omega-3 fatty acids. Fat is the most common component of brain nerve cells. Human physiology requires three omega-3 fatty acids, i.e., docosahexaenoic acid (DHA), alpha-linolenic acid (ALA) and eicosapentaenoic acid (EPA). Fish, eggs, and flax seeds are their most common natural sources [64]. PUFAs are essential for human health. The brain can generate neuronal signals in response to new experiences and stimuli. Neuronal plasticity, or the learning environment, is critical in long-term learning. DHA and omega-3 fatty acid levels must be balanced to maintain learning ability and enhance neuronal plasticity through membrane fluidity [64,65]. There is not much evidence to back up omega-3 supplementation’s effectiveness in improving the core or linked symptoms of ASD. Three randomized controlled trials (RCTs) comparing omega-3 fatty acids to a placebo revealed no significant differences [64,65]. The placebo group performed significantly better in one trial than the control group. Parent ratings of stereotypy and weariness in children who took omega-3 supplements against those who did not show substantial improvement after six months of treatment compared to the omega-3 group in externalizing behaviors [64,65]. It is advised to consult a physician before consuming omega-3 fatty acids as high doses can cause nausea, loose stools, and stomach upset.

### 3.2. Zinc for ASD

Zinc, a mood mineral, is vital because it is a cofactor for numerous neurotransmitters that affect mood and learning. Low zinc amounts disrupt dopamine production as this neurotransmitter involves learning and emotions such as motivation and pleasure [66]. A lack of zinc affects normal neural activities, including neurotransmission, brain development, and connection; moreover, it indirectly impacts the brain by impairing the immune system and changing the usual gut–brain link. This metal is essential for the neuropeptide social impact. So, to avoid autism, expecting and new mothers take a zinc supplement in their diets [67,68]. High doses of zinc can cause acute gastrointestinal symptoms like abdominal pain, diarrhea, and vomiting, so it is recommended to consult a physician to determine the dose. 

### 3.3. Vitamins for ASD

Most vitamins must be present in ideal amounts for healthy brain development. Vitamin D supplementation, in particular, has been demonstrated to help the symptoms of people with autism regress. Vitamins are potent antioxidants that help to protect cellular and mitochondrial function from free radical damage [66,68]. They also function as cofactors in a variety of biological processes. They regulate lipid and protein metabolism and are crucial for DNA synthesis. According to a study by Rollett, A. (1909), reduced folate levels during pregnancy are also related to congenital impairments. It has been connected to hyperactivity in youngsters. Autistic youngsters have also been proven to benefit from vitamin B1. Vitamin C has twin benefits, firstly as an antioxidant and secondly in creating some neurotransmitters [66,67]. Researchers have also determined that a specific gene encoding a particular protein is missing, which is the protein necessary to produce vitamin A. Clinical tests showed that vitamin A treatment enhances language and visual abilities in autistic patients. On the other hand, vitamin A supplementation must be carried out under the supervision of a physician as it can cause liver damage [68].

### 3.4. Iron for ASD

In autistic people, malabsorption of the vitamin inside the gastrointestinal system and their selective eating habits can lead to iron insufficiency. As a result, an iron shortage is reported to negatively affect sleep and neuroprotection. According to specific clinical investigations, cognitive impairment, reduced development, attention issues, and anemia are all related to mood swings in autistic children [69]. Children who have ASD have been found to have a high prevalence of iron deficiency (ID) and anemia that occurs due to iron deficiency (IDA). There are a small number of studies that link autistic clinical symptoms and iron deficiency indicators. The current research compares the levels of HB, hematocrit, Fe, ferritin, mean corpuscular volume, and red cell distribution width in patients with autism and healthy controls to determine the relationship between the numbers and symptoms. Children with ASD had lower HB levels than children without the disorder. Instead of the intensity of autistic symptoms, IDA in children with ASD may be linked to mental retardation [70]. A high dose of iron can cause iron poisoning, which shows multiple symptoms like nausea, abdominal pain, fever, headache, seizures, etc., so the dose needs to be advised by a physician to avoid complications due to a high dose of iron.

### 3.5. Magnesium (Mg) for ASD

Magnesium (Mg) works synergistically to relieve the clinical signs of autism. When autistic youngsters were given magnesium and vitamin B6, their social interaction and speech increased by 70% [71]. The most recognizable symptoms and indicators of Mg shortage are caused by neuronal and neuromuscular overactivity. In general, the connections between magnesium levels in inverse and direct sites and neurodevelopmental disorders may be a sign of higher excretion of magnesium in children with ASD, which ultimately results in a lower burden of magnesium in the body. The lack of noticeable changes in serum Mg levels may result from homeostatic regulation, which regulates absorption, excretion, and tissue redistribution (particularly in the bones) to maintain circulating Mg levels. Mg has a tremendous impact on neural excitation. Stress-related physical damage is more susceptible to Mg shortage, and Mg supplementation is protective. The neurologic impairment caused by experimental head trauma can be reduced pharmacologically by Mg, probably via the blockage of N-methyl-D-aspartate receptors. Mg salts help around 40% of people with autism when taken with large doses of pyridoxine, perhaps because they impact dopamine metabolism [72]. A high dose of Mg can cause diarrhea, vomiting, depression, low blood pressure, etc., so it is recommended to consult a physician before consuming Mg.

### 3.6. Selenium for ASD

Numerous vital metabolic processes for life depend on selenium. Countless studies have shown that the neuro-endocrine–immune network plays an important role in the interaction between the intestinal microbiota and the brain that impacts autism, and some animal studies have suggested that the gut microbiota may compete with the host for selenium when its accessibility in the organism becomes limited [73]. Selenium at high doses can cause nausea, bad breath, and fever as well as severe problems in the heart, liver, and kidneys, so it is advised to consult a physician before consuming selenium. 

## 4. Antioxidants for ASD

Antioxidant supplementation has been demonstrated to improve behavioral symptoms and reduce cognitive loss in patients with autism [74,75]. The following sections discuss several antioxidant substances that have demonstrated potential benefits for individuals with ASD. 

### 4.1. Curcumin for ASD

The ingredient in turmeric (Curcuma longa), sometimes known as “Indian Solid Gold”, is curcumin. It possesses anti-inflammatory and antioxidant properties and inhibits angiogenesis and cell adhesion. It also inhibits crucial cell signaling pathways, i.e., NFT-Kb and PI3K, indicating anticarcinogenic capabilities [74,75,76,77,78,79]. Curcumin’s neuroprotective effects make it useful in treating neurodegenerative illnesses, including Alzheimer’s Disease (AD), Huntington’s Disease (HD), Parkinson’s Disease (PD), and peripheral neuropathy [80,81,82]. Curcumin targets several cell signaling pathways, and its effects are as follows: increasing intracellular levels of glutathione, reducing inflammatory components, mitochondrial dysfunction, oxidative/nitrosative stress, and protein aggregation, counteracting the damage caused by heavy metals, and supporting liver detoxification. Its anti-proliferative impact on neurons is the principal method of preventing brain-stimulated microglia and reactive astrocytes from releasing cytokines and other active elements [83,84]. Shu-Juan et al. 2012 [85] showed that curcumin has neurotherapeutic potential by improving autistic behavior and boosting brain-derived neurotrophic factor (BDNF) levels in sodium valproate rat models of autism. For two weeks, 35-day-old rat pups were administered a 10 g/L concentration of curcumin. Their social interactions improved significantly and repetitive behavior decreased, and there were increased BDNF levels in the temporal brain [84,85,86,87,88,89,90]. Still, the role of curcumin in autistic phenotypes remains unclear. 

### 4.2. Resveratrol for ASD

Resveratrol is a phenolic acid stilbenoid produced when bacteria and fungi attack plants. It can be found in berries, grapes, and almonds. Resveratrol is effective against oxidative stress and immune function and has the potential to cross the blood–brain barrier (BBB). Resveratrol interacts with a variety of targets, is multifactorial in nature, and inhibits cyclooxygenase (COX), activates sirtuin (silent mating type information regulation homolog 1—SIRT1), induces endothelial nitric oxide synthase (eNOS), and activates peroxisome proliferator-activated receptors (PPARs) [91,92,93,94,95]. Resveratrol allosterically modulates the regulatory target SIRT1. It promotes AMP-activated protein kinase (AMPK) phosphorylation and decreases oxidative damage in F2 hybrid mice [94,95]. Fontes-Dutra et al. 2018 [96] looked at the neurotherapeutic potential of resveratrol (RSV) in a valproic acid (VPA) animal model of autism. The study’s primary purpose was to see the neurodevelopmental abnormalities that might be caused by prenatal valproic acid exposure and whether resveratrol could be utilized as a treatment [96]. The effects of resveratrol on sensory behavior were investigated after autism was induced in rats. The location of GABAergic parvalbumin (PVC) neurons in sensory brain areas and the expressions of excitatory and inhibitory synapses were studied [97,98] in rats with an ASD-like phenotype produced by propionic acid (PPA). Resveratrol was administered at 5, 10, and 15 mg/kg [99]. The therapy started the day following the operation and lasted for 28 days. Rats were subjected to behavioral tests between the 7th and the 28th days. Sociability, repetitive conduct, anxiety, unhappiness, and item recognition tests and the Morris water maze test for perseverance were some of the behavioral tests used [100]. They discovered that matrix metalloproteinase-9 (MMP-9) activation caused mitochondrial dysfunction and the production of inflammatory cytokines. Resveratrol restored the core and associated symptoms of autistic phenotypes by suppressing oxidative–nitrosative stress, mitochondrial dysfunction, and TNF-α and MMP-9 expression. Based on their findings, we can say that resveratrol can be a promising therapeutic intervention for the management of ASD [101,102,103].

### 4.3. Naringenin for ASD

Flavanone Naringenin (NAR) is abundant in grapefruit, oranges, and tomato skin [104]. Naringenin inhibits human cytochrome P450-metabolizing enzymes of the CYP1A2 isoform [105,106]. Naringenin has an antioxidant effect by inhibiting the NF-κB pathway, which reduces oxidative injury caused by radiation exposure in mice. It also has an antihyperlipidemic effect by preventing the secretion of very low-density lipoproteins (VLDL) [106,107]. BDNF signaling also has antidepressant potential in chronic unpredictable mild stress. Because of its ability to inhibit cell proliferation by binding to estrogen receptors, it has versatile functions in many cancers [108]. It is also beneficial in treating osteoporosis, cancer, and cardiovascular diseases. Naringenin also suppresses neuroinflammation in glial cells by triggering the suppressor of cytokine signaling 3 (SOCS3)-3. Because the NF-B pathway is inactivated, it had a neuroprotective role in a middle cerebral artery occlusion (MCAO) model of ischemic stroke (IS) [109,110,111]. Naringenin works on ASD in a similar manner as it works on IS. In their study, Bhandari et al. (2018) recently looked at the neurotherapeutic potential of naringenin, naringenin-loaded glutathione, and Tween-80-coated nanocarriers in treating ASD [112]. They helped to reverse the neuropathology that had developed. PPA administration improved brain uptake by avoiding naringenin’s low oral bioavailability and at a low oral dose of 25 mg/kg. As a result, these brain-targeted naringenin nanocarriers can be used in clinics as a neurotherapeutic [113,114]. Based on previous findings, we can suggest that naringenin can be a promising non-pharmacological therapeutic approach to the management of ASD.

### 4.4. Sulforaphane for ASD

Sulforaphane is a phytochemical belonging to the isothiocyanate group and is chemically recognized as 1-isothiocyanate-4-(methylsulfonyl) butane. The photocatalytic activity of myrosinase produces sulforaphane, once glucoraphanin is metabolized [115,116]. Glucoraphanin is a precursor to sulforaphane and can be found in various vegetables. Broccoli and cauliflower belong to the cruciferous family. According to Yuesheng Zhang [117], sulforaphane is beneficial for reducing oxidative stress in the human body, reducing mitochondrial dysfunction. It is a neuroprotective compound that prevents apoptosis in hippocampal neurons due to oxidative stress and the production of free radicals. It also has anti-diabetic properties [118,119]. It has anticarcinogenic and anti-inflammatory properties, aiding in lowering the infarct volume after ischemic stroke. It works by activating both the nuclear factor erythroid 2-related factor 2 (Nrf2)-dependent and the Nrf2-independent self-contained pathways [120,121]. Astrocytes activate the Nrf2 response, and heat shock protein 27 is upregulated [122]. Singh et al. 2014 [123] investigated the neurotherapeutic effects of sulforaphane in young men aged 13 to 27 with moderate-to-severe ASD. This was a placebo-controlled, randomized, double-blind clinical trial. The patients were given sulforaphane at a 50–150 mol/day dose for 18 weeks, and the results were monitored during a four-week drug-free period [124]. There were 29 ASD patients and 15 control subjects in a placebo-controlled study. After 18 weeks of treatment, their behavior was assessed using behavioral rating scales. According to the findings, treatment with sulforaphane improved the patients’ social interaction abilities and reduced the deficits overall [124]. This was demonstrated by an increase in behavioral assessment scores using behavioral rating scales such as the Aberrant Clinical Global Assessment (CGA), Autism Behavior Checklist (ABC), Social Responsiveness Scale (SRS), and Clinical Global Impression—Improvement (CGI-I). As a result, broccoli’s sulforaphane can decrease oxidative stress neuroinflammation and prevent DNA damage [124]. Sulforaphane appears to be a safe and effective management option for ASD and other neurological disorders.

### 4.5. Luteolin for ASD

Luteolin is a natural flavonoid with anti-inflammatory, anti-antioxidant, and neuroprotective properties and can easily penetrate the BBB due to its low lipophilicity. Luteolin has the potential to neutralize ROS and downregulates IL-1beta, IL-6, and TNF-alpha, which might counteract neuroinflammation. It also inhibits the stimulation of astrocytes, as well as microglial activation and proliferation [125]. In a mouse model, luteolin inhibited IL-6 release from activated microglia and reduced maternal IL-6-induced autism-like behavioral deficits related to social interaction [125]. In a recent study by Marianna et al. 2022, chronic luteolin treatment ameliorated hyperactivity, memory, and motor skills in 3dkL5 +/− mice by inhibiting neuroinflammation [126]. Even luteolin shows positive results in clinical models, which is discussed in Table 1. Based on the preclinical and clinical data, luteolin can be an excellent natural medicine for the management of autism. 

## 5. Camel Milk for ASD

Camel milk may have recently been used to treat several illnesses, including food allergies, diabetes, hepatitis B, autism, and other autoimmune diseases. In patients suffering from reduced plasma glutathione peroxidase (GSH-Px), superoxide dismutase (SOD) and cysteine were linked to ASD, and the effect of camel milk was documented. It showed improvement in ASD clinical outcomes [127,128]. Camel milk has more essential nutrients, like Ca, Fe, Mg, Cu, Zn, K, vitamin A, vitamin B2, vitamin C, and vitamin E, than the milk of other herbivorous animals. Moreover, camel milk lacks beta-lactoglobulin and beta-casein, two vital active ingredients that are components of cows’ milk and cause milk allergies [129]. Camel milk includes different preventive biomolecules with antimicrobial, anti-viral, and immunologic characteristics. It contains anti-inflammatory protein molecules and antibodies that aid in easing specific primary autistic symptoms [130]. Those antibodies have new structural features with better tissue penetration and hidden epitopes. These characteristics may help avoid infections and provide potential benefits [131]. Furthermore, the nanobody structure of camel milk is highly comparable to the antibodies of immunoglobulins from humans (IgG3) [131]. This implies that the antibodies from camels are similar to the antibodies from humans. Camel milk has a unique composition that allows it to be used in various ways. Improvements in children with autism have been shown to occur. ASD can be treated by increasing superoxide dismutase levels and Plasma GSH, as well as by lowering oxidative stress, which is a component of the etiology of autism [127,128]. According to Gader et al. 2016 [132], camel milk reduces cancer risk. Symptoms of autism have improved significantly, or there has been a significant improvement in basic skills. A study by Al-Awadhi LY et al. (2015) suggests that the antioxidant enzymes and non-enzymatic antioxidant molecules found in camel milk could help to improve typical ASD behaviors [133]. Large-scale dose-focused investigations are necessary to verify the impact of camel milk on oxidative stress parameters and the therapy of autism [133]. Camel milk could be a promising therapeutic intervention for the management of ASD.

## 6. Hormone Therapies for ASD

Children and men with autism have improved social interaction and speech following hormone therapy [134]. Below we discuss in detail studies related to different hormone and the results of hormone therapies using melatonin, oxytocin, and vasopressin.

### 6.1. Melatonin for ASD

Melatonin use for curing sleep disturbances in children with ASD is supported by research. A recent significant randomized, double-blind placebo control (RDBPC) study found that pediatric-appropriate prolonged-release melatonin mini tablets (PedPRM) enhanced bedtime and quality of sleep, with an enhancement in total sleep time and sleep quality [134,135,136,137,138]. Finally, latency and sleep disturbances were reduced. Aside from its sleep-related benefits, a few RDBPC studies have shown that melatonin has been shown to improve communication, rigidity, and mood and reduces anxiety and depression in children with ASD [134,135,136,137,138]. Melatonin could be a promising hormone therapy for the management of ASD. 

### 6.2. Oxytocin for ASD

According to new evidence, neuropeptides like oxytocin could be helpful in treating core ASD symptoms. According to a recent meta-analysis, oxytocin does affect children’s social cognition and restricted and repetitive behaviors (RRBs) in autism spectrum disorder (ASD). The results of an unreviewed RDBPC study of oxytocin levels in students with autism were presented during an oral exam presentation at the International Autism Conference [139,140]. According to a 2017 International Meeting for Autism Research (IMFAR) study, oxytocin was not superior to other hormones compared to a placebo in reducing social withdrawal. However, it outperformed the placebo in improving social recognition. In contrast to the placebo, oxytocin improved social functioning as evaluated using the Social Responsiveness Scale (SRS) in a recent study of children with ASD RDBPC [139,141]. Based on previous findings, oxytocin could be a promising hormone therapy for the management of ASD.

### 6.3. Vasopressin for ASD

Vasopressin’s improved reactions to personal interaction and communication and proof from preclinical studies recommend that Vasopressin receptor 1A (V1AR) antagonists may have pro-social advantages for disorders in which emotional and social functions are core deficits, such as schizophrenia [142]. Antidepressant and anxiolytic properties are also present. The safety and effectiveness of V1a antagonists in ASD have been studied in a few clinical trials [142]. In one study, the vasopressin V1a antagonist RG7713 was given intravenously to adults with high-functioning ASD to enhance social communication. Balovaptan (RG7314), another vasopressin V1a antagonist, has shown promise in improving communication and social interaction in ASD patients. The Food and Drug Administration (FDA) recently granted PB2452 status as one such compound with “Breakthrough Therapy Designation”, raising hopes for approval of the first pharmacological method to enhance core social communication deficits in ASD. Vasopressin can cause multiple adverse effects like allergic reactions, stomach pain, nausea, irregular heartbeats, low blood sodium levels, low blood pressure, and pale skin [140,141,142]. Based on previous findings, we can suggest that vasopressin could be a promising hormone therapy for the management of ASD.

## 7. Herbal Medicine for ASD

A large variety of herbal remedies, which include *Gingko Biloba*, *Zingiber officinale* (ginger), *Astralagus membranaceous*, *Centella Asiatica* (Gotu cola), and *Acronis Calamus* (Calamus), may have therapeutic benefits in ASD patients due to their somatic effects such as increasing cerebral blood flow circulation, cognitive function enhancement, soothing or sedative effects, and enhancing the immune system’s response [143,144]. A recent comprehensive review found that herbal remedies are safe when used in conjunction with conventional therapy and have positive benefits in controlling the aberrant behaviors and inattentiveness of ASD patients [143,144]. However, such results should be interpreted with caution due to the dearth and ambiguity of available data. Additional limitations on how these treatments can be used have been created due to risks associated with herbal interactions that may arise with other drugs and the questionable sources of herbs. Additional clinical trials are required to advance the research and validate Ayurvedic remedies’ potential therapeutic benefits in ASD [143,144]. 

## 8. Cannabidiols for ASD

Cannabidiol (CBD) is a phytocannabinoid in the cannabis plant *Cannabis sativa* (marijuana). Although marijuana contains hundreds of phytocannabinoids, CBD is the second most prevalent after delta-9-tetrahydrocannabinol (THC), which has psychoactive characteristics [145]. Marijuana has been used for fiber and therapeutic purposes in India, China, and the Middle East for over 8000 years. It was spread to Europe by Napoleon’s soldiers arriving from Egypt in the 1800s, and a doctor who campaigned in India subsequently brought it to Britain for medicinal use [145,146]. When used sufficiently, tetrahydrocannabinol (THC), phytocannabinoids’ principal psychoactive component in *Cannabis sativa*, can aggravate various neurological conditions. Cannabidiol (CBD) may also help to decrease autistic behavior. This chemical offers therapeutic benefits such as immunomodulation, antioxidant defense, and neuroprotection with few or no adverse effects [147,148]. Sleeplessness, tension, discomfort, and even motor deficits like PD trembling are among the medical diseases that are treated using the non-psychoactive component of marijuana known as CBD. Some of the mechanisms through which BD exerts its neuromodulatory and neuroprotective effects include agonist potentiation, oxidative activity enhancement, 5HT1A transmitter engagement, and anandamide level augmentation. [148,149,150,151]. The endocannabinoid system (ECS) contains CB1 in the central nervous system (CNS) and CB2 throughout the body and immune system. The ECS regulates cognition and behavior by modulating synaptic transmission across the CNS. The ECS consists of endocannabinoids, cannabinoid receptors, and enzymes for synthesizing and degrading these endocannabinoids. Due to neurological signaling, endogenous cannabinoids are created and liberated from the phospholipid bilayer attached to the postsynaptic barrier [148,149,150,151]. By activating cannabinoid sensors on the neuromuscular junction and preventing transmitter release from the presynaptic cell, they function backward, signaling molecules across the synaptic gap. Phospholipase C and diacylglycerol lipase (DAGL) are two enzymes involved in synthesizing 2-arachidonoylglycerol (2-AG) [151]. Additionally, it has been revealed that cannabidiol acts as a positive allosteric modulator at GABAA receptors, and observational trials have shown that CBD in the formulation of Epidiolex is indeed an effective analgesic in the treatment of Lennox–Gastaut disorder and Dravet syndrome [152,153]. By controlling the balance of interneurons transmission, CBD’s potential to increase endogenous cannabinoid production levels and promote the GABAergic transfer of information may aid in restoring neuronal function and neuroplasticity [152,153]. In ASD and Fragile X Syndrome (FXS), in which patients do not have seizures, CBD treatment has been proven to help in both animal and human models. Cannabidavarin (CBDV), also a chemical in the Cannabis sativa plant, is now being studied in animal models of ASD. Recently, discovering a THC-free topical CBD has allowed for their investigation in both ASD and FXS [154,155]. Hessler et al. 2019 [156] undertook an accessible trial in Australia utilizing dermal cannabidiol at a dose of 250 mg bi-daily, for twelve weeks, on children with FXS aged 3–17 years old. ZYN002 is a patent-protected clear-gel pharmaceutically manufactured synthetic cannabidiol with a permeation-enhanced formulation for effective transdermal delivery. The significant endpoint, the Stress, Distress, and Emotion (ADAMS) scale, and the secondary measures, including the ABC, exhibited effectiveness [157,158,159,160]. In conclusion, a nationwide research experiment including almost 200 children with FXS was carried out, and the significant consequence predictor was only demonstrated by children with FXS who had more than 90% methylation using the ABC and the FX Social avoidance subscale, a scale designed for FXS and adapted from the ABC [159]. Zyn002 is now undergoing a second international experimental investigation to obtain FDA approval for general use, although it still needs this approval. There is a good chance that the upcoming observational trials for autism and FXS will assist some subpopulations, including both diseases, and that marijuana use might increase [156,160]. Based on previous findings, cannabinoids could be a promising therapeutic intervention for the management of ASD. Table 1 represents the names of natural products and clinical data related to autistic children and adults. 

**Table 1 medicina-59-01584-t001:** Published clinical data on different nutritional therapies for management of autistic children and adults.

Name of Natural Product	Clinical Model of ASD	Method and Duration	Result	References
Cannabidiol	150 participants (5–21 years of age)	Entire plant cannabis extricate that included cannabidiol and \sΔ9-tetrahydrocannabinol at a 20:1 ratio and distilled cannabidiol and Δ9-tetrahydrocannabinol at a 20:1 ratio.	Whole-plant extract showed 49% improvement in behavior with no severe effect, with common adverse effects like decreased appetite and somnolence.	[146]
Cannabidiol	18 autistic patients	Observational study	Cannabidiol and enriched *Cannabis* *sativa* extract decreased multiple ASD symptoms, even in epileptic patients	[159]
Luteolin	Children (*n* = 37, 4–14 years old)	Children were given luteolin as well as another supplement for four months	50% improvement in eye contact and attention, 25% improvement in social skills, 10% improvement in speech, and 75% improvement in GI	[161]
Luteolin	10-year-old male child	Co-ultra peak-LUT	Improved almost all symptoms	[162]
Luteolin	Children	Based on serum levels of IL-6 and TNF	Reduction in IL-6 and TNF levels after 26 weeks of treatment improved behavior	[163]
Luteolin	50 children aged 4–10 years old (42 boys, 8 girls)	Open-label trial, one capsule per 10 kg of weight per day with food	Decreased all clinical signs with no significant harmful effects	[164]
Cannabidiol	34 healthy men (half with ASD), 600 mg cannabidiol taken via oral administration	fMRI response to cannabidiol in ASD	Cannabidiol altered the fractional amplitude of low-frequency fluctuations.	[165]
Cannabidiol	34 healthy men (17 neurotypical and 17 ASD)	A single oral dose of 600 mg cannabidiol or placebo	Modulated glutamate GABA system	[166]
Cannabidiol	188 ASD patients	Medical cannabis treatment	28 patients showed significant improvement, 50 moderate improvement, 6 slight improvement, and 8 no improvement	[167]
*Ginkgo biloba* extract (Ginko T.D., Tolidaru, Iran)	3 autistic patients	2 × 100 mg, four weeks	Improvement in behavior	[168]
Camel milk	Total of 45 children, three groups of 15 children each	Blood samples for activation-regulated chemokine (TARC) serum level and childhood autism rating scale (CARS) score were taken before and after participants consumed 500 mL of milk per day in their daily diet for two weeks.	Reduced level of TARC and improvement in CAR score	[169]
Gluten-free diet	80 children, two groups (one regular group consisting of 40 children and one gluten-free diet group consisting of 40 children), and 53.9% of children had gastrointestinal abnormalities	The Rome questionnaire was used to examine gastrointestinal symptoms, and the Gilliam Autism Rating Scale 2 (GARS-2) was used to assess psychometric qualities.	Reduction in gastrointestinal symptoms and ASD symptoms	[170]
GFCF diet	37 patients, six months on a regular diet and six months on GFCF	Questionnaires regarding behavior	No change in behavior after consumption of GFCF for 6 months	[171]
GFCF diet	14 children, 3–5 years age	12-week double-blind, placebo-controlled trial study with continuation of the diet, with a 12-week follow-up and dietary supplement delivered via snacks	No change in behavior or other autism symptoms	[172]
Modified ketogenic, gluten-free diet	15 children, 2–17 years of age	Open-label clinical trial for three months	Improvement in autism symptoms	[173]
Vitamin and omega 3	111 children	Trial: Vitamin D (2000 IU/day), omega-3 LCPUFA (722 mg/day EPA and DHA, OM), or both for 12 months.	Vitamin D and omega-3 LCPUFA reduced irritability symptoms	[174]

## 9. Relationship between Microbiome and ASD

Our bodies contain several hundred million microbial colonies that code a hundred times the additional genetic traits of human genetics, including the most recent update predicting a microbial proportion of 1.3 species per individual cell, down from the highly cited 10:1 and 100:1 ratio [175]. With the massive microbial spread, the human body’s microbiome can play therapeutic and pathogenic roles [176,177]. Animal models and human subjects have been used to study microbiomes and ASD. The maternal influence on early intestine development is crucial. Maternal colonization in the offspring is frequently influenced by natural factors [176,177]. The microbiota’s perinatal and prenatal periods have an impact on the microbial makeup of the uterus. The olfactory system mediates the spread of harmful oral germs via ectopic translocation. BBB disruption and perivascular and circumventricular space neuroinflammation may be caused due to an oxidative disorder in the brain, implying that the oral microbiota and dysbiosis impact the brain [178,179]. Another possible pathway is believed to be the gastrointestinal system and nervous axis, which regulate the oropharynx. It has a crucial function in ASD pathology. In general, the interaction is positive. A complex mechanism exists between the microflora and the nervous system in autism. Short-chain fatty acids (SCFAs) and BDNFs, among other natural and hereditary factors, may explain, control, and modulate epigenetic pathways. The oropharynx, which plays a vital role in the pathophysiology of the buccal space, is thought to be a mediator in ASD [180,181,182,183]. Figure 2 explains the relationships between different microbiomes and ASD.

### 9.1. Oral Microbiome for ASD

According to the theory, microbial perturbations in the stomach might migrate to the oropharynx and affect the oral microbiome [184]. Children with ASD usually have speech issues and are incredibly picky eaters. Therefore, the oral microbiota was identified as a possible diagnostic sign for ASD [185]. The oropharynx is one of the essential parts of the digestive system. Five sensory motor cranial nerves connect it to the rest of the body and link it to the GI tract. It is thought that the development of autism may be significantly influenced by a potential exchange mechanism within the brain [186]. The gut–brain axis allows for a connection between the gut and the brain, confirming this theory. Oral bacteria enter the brain, causing responses like inflammation, metabolism disturbance, and spinal cord infection [187]. The olfactory nerve is supposed to work as a sensor in the olfactory tract. The bacterial dispersion to the brain via blood circumventricular organs or perivascular spaces is thought to be mediated by a damaged BBB. *Haemophilus parainfluenza*, a Gram-negative bacterium, and its metabolites have been linked to oral diseases. Even routine dental procedures can cause bacteremia, and a portion of these microbes may traverse the BBB. Altered transcript expression has been described in the microglia of ASD individuals, and disrupted microglia function could impair BBB integrity. This could expose the brain to bacterial metabolites, thereby triggering an inflammatory response and altering metabolic activity within the central nervous system. The prolonged disruption of energy metabolism within neurons, oligodendrocytes, and glia could lead to structural changes in the cortex, hippocampus, amygdala, or cerebellum, which have all been documented in ASD individuals to be increased. They could penetrate the BBB, harm the nervous system, and cause ASD. This could expose the brain to bacterial metabolites, thereby triggering an inflammatory response and altering metabolic activity within the central nervous system. Gram-negative, putative periodontal pathogens, are rich in lipopolysaccharides (LPS), which exhibit pro-inflammatory activity. The leakage of LPS through the BBB in ASD individuals could lead to inflammation in the central nervous system (CNS) [188,189,190,191,192]. The microorganisms seen in patients with periodontal disease are not common in healthy people. Numerous health problems have been linked to periodontal disease [193]. The risk of early birth increases by 2–7 times since the organisms that cause intrauterine infections have been found in the mouth rather than the urogenital tract. [193,194,195]. Placentae have been shown to indicate the mouth microbiota more than the vaginal microbiome, signifying hematogenous spread, especially in underlying periodontal disease and oral intercourse [196]. Such colonization may lead to infection inside the uterus. Eighty-five percent of the oral microbiota is introduced in the initial six months after birth in the newborn–early childhood period, and children’s faces resemble their mothers [196,197,198].

### 9.2. Gut Microbiome for ASD

The most well-researched aspect of autism is the intestinal flora. Bacteria in a higher organism are found in the gastrointestinal tract, and they play major physiological roles in metabolic activity, digestive health, immune function, and endocrine and neurological function. Any imbalance in the relationship between the intestinal bacteria and the different human cells might result in sickness. Microbes impact the host’s vital biological processes and may be a significant factor in the etiology of many diseases [178,179,199,200]. The importance of gut bacteria in public health has prompted studies to emphasize the significance of identifying these microbes as potential contributors to the development of ASD. *Prevotella*, a promising health biomarker, more abundant in neurotypical people but almost non-existent in autistic people, is another notable Gram-negative bacteria genus [201,202,203]. *Prevotella* is abundant in people who eat a diet high in phytonutrients, complex carbs, and the oil obtained from fish, and is essential for normal brain development. It metabolizes energy to develop vitamin B1, which helps with the signs of autism [204]. In controlled and prospective clinical studies, two forms of vitamin B12 have been investigated: (1) There is evidence that subcutaneously administered mB12 improves methylation and the clinical symptoms of ASD. Redox metabolism also seems to be linked to improvements in clinical symptoms, biochemistry, and physical medical diseases, particularly in people with unfavorable biochemistry and when paired with folinic acid (also known as leucovorin). (2) A combination of cB12 and mB12 contained in an MVI. *Prevotella* deficiency suggests distinct nutritional autism-related habits in children that alter gut microorganism combinations and influence neurodevelopment, implying that restoring it could be therapeutic [205]. Among the Firmicutes phyla is the Gram-positive genus *Clostridium*, which is more prevalent in ASD patients. The associated species are *Clostridium boltae* and *Clostridium histolyticum*. These Gram-positive bacteria produce enterotoxins, which cause diarrhea by damaging intestinal tissue. They may also contribute to the increased cellular uptake of heavy substances like protein obtained from cereals and milk [206]. Moreover, the advantageous bacterial population of a *Bifidobacterium* was discovered in lower numbers among those with autism. Intestinal dysregulation in people with ASD has been confirmed through a rise in potentially pathogenic microbes and a reduction in beneficial bacteria [205,206]. In a different light, ASD patients’ apparent lack of gut microbe variety and prosperity increases their susceptibility to a vulnerable gut environment, resulting in GI disturbances, infections, and autistic behaviors [204]. Finally, a symbolic change in the gut’s microbial composition disrupts critical physiological processes, which affects the behavioral manifestations of autism and further leads to an utter lack of favorable microbiological byproducts, the release of toxic microbiological lipopolysaccharides, and the pathogenic incursion of the gut lining by immunological cytokines that promote neuroimmune inflammation [205,206].

### 9.3. Vaginal Microbiome for ASD

The mother’s characteristics that influence the growth of autism are referred to as vaginal microbiota. Because humans are born germ-free, it is assumed that the first colonization of bacteria in the human stomach occurs during birth and travels through the vaginal opening. However, evidence is emerging that colonization of the uterus may begin much earlier [207,208,209,210,211,212,213]. Contamination happens during a cesarean delivery when the infant touches the maternal epidermis and other pathogens. The microbial population that is transferred via vertical transmission impacts the microbial community in the intestine [214,215,216]. As a result, the substantial disruption to the microorganisms in the mother’s genitalia due to high metabolic demand unintentionally impairs neurodevelopment in infants at a critical stage in brain development [217]. The vaginal canal contains more than fifty bacterial species, with *Lactobacillus* being the most common in healthy women [218]. It has been shown that maternal anxiety during the first trimester seems to have an inhibitory effect on genital immunology and the proportion of bacteria responsible for the production of lactose, which causes an imbalance of microbes in the gut. The imbalance of gut microbes is transmitted vertically to children. Vaginosis caused by bacteria is a common ailment. According to research, premature toddlers have a 10-times higher risk of developing ASD than those born at full term. According to research trials and extensive epidemiological studies, many maternal prenatal infectious diseases and the increased signaling molecules produced by immune cells increase the chance of autism in children [219,220,221,222].

## 10. Conclusions

ASD has multiple causes, including epigenetic changes and complex genetic mutations, and due to these complications, it is hard to manage ASD. Currently, to manage ASD symptoms, risperidone and arpiprazole are the only two drugs approved by the FDA. There is ambiguity around this illness because there are now just a few alternatives for therapeutic intervention. In this review, we shed light on the non-pharmacological interventions that can help to manage the symptoms of ASD. This review includes all the nutritional approaches, including dietary phytochemicals like curcumin, resveratrol, naringenin, sulforaphane, camel milk, herbal medicines, different hormones, and microbiomes, for managing ASD. These are possible safe and efficient approaches to lessening the burden of the disease and are important because healthcare is expensive, and there is a significant burden on ASD carers. There is an urgent need for multidisciplinary research focusing on drug delivery techniques in the brain for the above-mentioned dietary supplement and microbiome therapy. All non-pharmacological interventions need rigorous clinical trials to bring them to the market, but a few of them can be consumed in their current state with FDA-approved drugs to manage ASD symptoms.

## Figures and Tables

**Figure 1 medicina-59-01584-f001:**
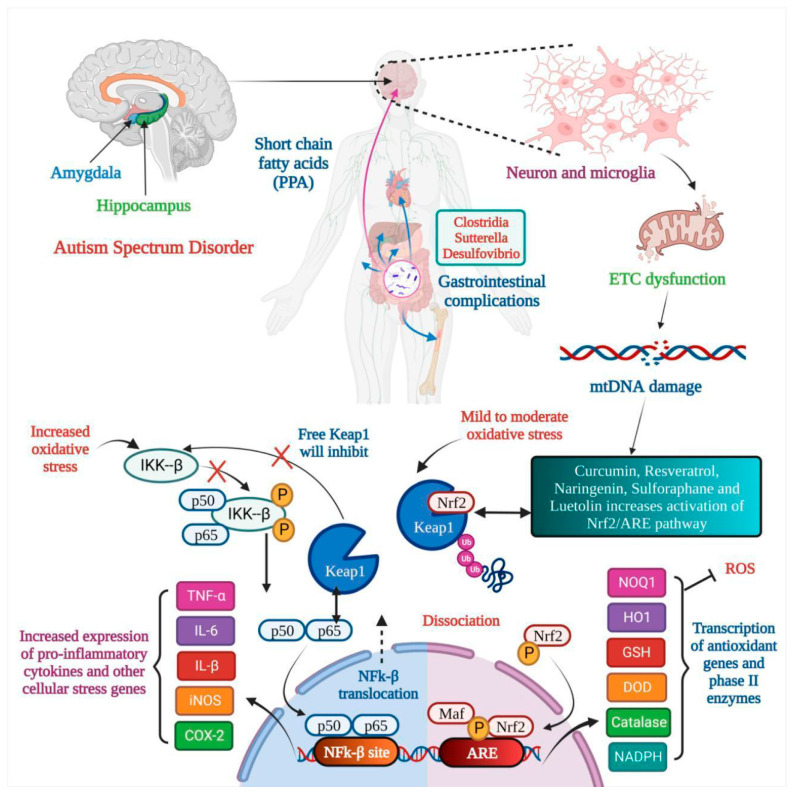
Nutritional therapy attunes mitochondrial dysfunction in Autism Spectrum Disorder. In autism, the amygdala and hippocampus are affected, and gastrointestinal complications are seen. Microglia activation leads to electron transport chain (ETC) dysfunction, which results in mtDNA damage, and oxidative stress leads to dysfunction of the Nrf2 pathway and natural products like curcumin, resveratrol, etc., resulting in inactivation of the Nrf2/ARE pathway. Two outcomes are seen: (a) the transcription of antioxidant genes and (b) the increased expression of pro-inflammatory cytokines.

**Figure 2 medicina-59-01584-f002:**
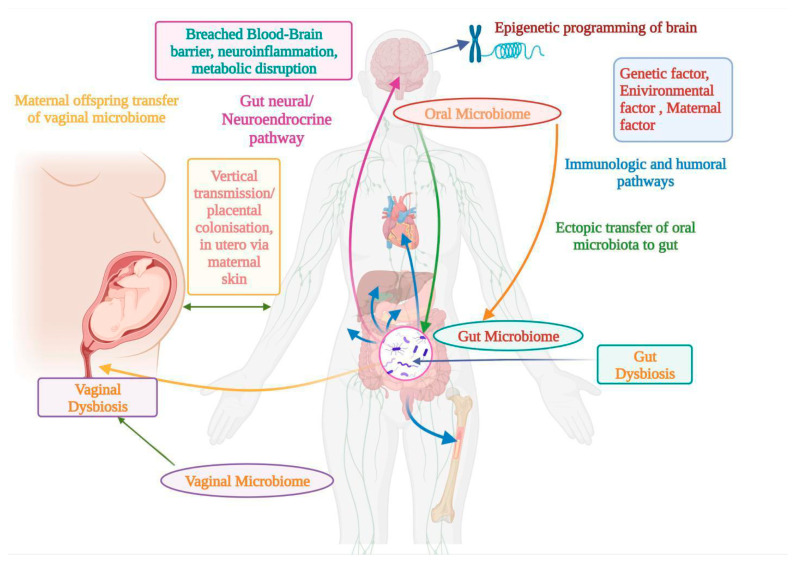
Linkage between ASD and oral, gut, and vaginal microbiomes, along with different factors affecting pathways and their effects on neuroinflammation and metabolic disruption.

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
