# Peer review of "A Spectrum of Solutions: Unveiling Non-Pharmacological Approaches to Manage Autism Spectrum Disorder"

_medicina, 2023, doi:10.3390/medicina59091584_

Round 1

Reviewer 1 Report

The manuscript provided a good overview of the history and current understanding of autism spectrum disorder (ASD), as well as the challenges in treating and managing the condition. However, there are a few areas where the manuscript could be improved:

1.     The introduction could provide more context and background information on the prevalence and impact of ASD. For example, it would be helpful to include information on the economic and social costs of ASD, as well as the impact on families and caregivers.

2.     The last sentence of the introduction states that the review will "open a new horizon for the treatment and management of ASD," but it is not clear how this will be achieved or what the implications of the review's findings will be. Further elaboration required to strengthen the contribution of the present manuscript to the field. 

3.     The statement that "Behavioural and pharmacological therapeutic approaches for ASD are somewhat ineffective" (on page 2) is a sweeping generalization that is not supported by the evidence. While these approaches may not work for all individuals with ASD, they have been shown to be effective for many. It would be more accurate to say that there is a need for additional treatment options that can improve outcomes for individuals with ASD.

4.     The content could benefit from a more critical analysis of the evidence for each dietary approach. While some studies have shown promising results for certain diets, others have not found any significant effects. It is important to note the limitations of the research, including small sample sizes, lack of control groups, and variability in study design. Additionally, some of the claims made in the review, such as the link between casein and gluten and "excess opioid" theory, are controversial and not universally accepted in the scientific community.

5.     The content could be improved by including more information on the potential risks and drawbacks of each dietary approach. For example, the elimination diet can lead to malnutrition if not carefully monitored, while the specific carbohydrate diet may be challenging to follow and restricts certain foods that are important for overall health. Additionally, some nutritional supplements may interact with medications or have harmful side effects if taken in excessive amounts.

6.     It would be helpful to provide more information on the specific symptoms that each nutrient is believed to target in individuals with ASD. For example, while the review mentions that magnesium and vitamin B6 supplementation has been linked to improvements in social interaction and speech, it would be useful to provide more detail on the specific behavioral and cognitive domains that have been shown to improve with these nutrients.

7.     Although the review mentions some studies that have investigated the effects of various nutrients on ASD symptoms, it does not provide a critical evaluation of the quality of these studies or the strength of the evidence they provide. It would be helpful to discuss the limitations of these studies. 

8.     While the review mentions the potential benefits of curcumin, resveratrol, and naringenin for individuals with ASD, it does not provide a thorough discussion of the underlying mechanisms of action through which these antioxidants may exert their effects. It would be helpful to provide more detail on the specific neuroprotective and anti-inflammatory effects of these antioxidants and how these effects may be relevant to the management of ASD symptoms.

9.     The review focuses primarily on studies conducted in animal models of ASD, rather than in human subjects. While animal studies can provide valuable insights into the potential therapeutic effects of antioxidants, it is important to also discuss the results of human clinical trials, if available. This would provide a more accurate picture of the potential benefits of these antioxidants for individuals with ASD.

10.  While the review briefly mentions the underlying mechanisms of action of sulforaphane and luteolin, it could benefit from a more detailed analysis of the specific pathways through which these compounds exert their effects on ASD symptoms. This would provide a more nuanced understanding of the potential therapeutic benefits of these compounds for individuals with ASD.

11.  While the review discusses the results of several studies on the use of melatonin for sleep disturbances in individuals with ASD, it could benefit from a more detailed analysis of the underlying mechanisms through which melatonin exerts its effects on ASD symptoms beyond sleep. This would provide a more nuanced understanding of the potential therapeutic benefits of melatonin for individuals with ASD.

12.  While the review briefly discusses the potential therapeutic benefits of vasopressin for individuals with ASD, it could benefit from a more detailed analysis of the underlying mechanisms through which vasopressin exerts its effects on ASD symptoms. Additionally, the review could discuss any potential risks or adverse effects associated with vasopressin treatment, especially given that individuals with ASD may be more susceptible to adverse reactions to medications.

13.  While the review briefly touches on some potential pathways, such as the gut-brain axis and the impact of oral bacteria on the brain, it could benefit from a more in-depth discussion of the specific biological mechanisms that may be involved.

14.  The review could benefit from a more detailed discussion of the potential implications of the findings for ASD diagnosis and treatment.

15.  While some of the interventions mentioned in the conclusion may have potential benefits, there is limited empirical evidence to support their effectiveness in treating ASD. Many of the studies on these interventions have small sample sizes and lack rigorous design, making it challenging to draw definitive conclusions about their efficacy.

16.  The conclusion states that there is an urgent need for multidisciplinary research focusing on drug delivery techniques in the brain for dietary and microbiota therapy. While such research is undoubtedly important, the conclusion overlooks the fact that pharmacological interventions, such as medications for anxiety or attention-deficit/hyperactivity disorder, are often an essential part of managing ASD for many individuals. The conclusion suggests that non-pharmacological interventions should be the primary approach to managing ASD, which oversimplifies the complexity of the disorder and the range of interventions needed to address it.

Regarding the writing style, the manuscript contains several long sentences that could be broken up into smaller ones for easier readability and comprehension.

Author Response

We would like to thanks the reviewer for spending the valuable time in reviewing the manuscript and sharing the valuable comments. As per the suggestion the manuscript is revised and modifications are shown in blue color. All the comments has been addressed and report is attached. 

Reviewer 2 Report

Thank you very much for giving me the opportunity to review this very interesting manuscript. With some improvements, the manuscript has the potential to be educational (for example. University students), informative, and comprehensive in the field of Autism Spectrum Disorder. Overall the manuscript is generally well-written, organized, and appropriate for the scope of the Journal. The writing style is appropriate for a scientific review and is very helpful as it consists of many figures. Although there are some suggestions:

The review highlights the potential advantages of these interventions in effectively managing symptoms associated with ASD and enhancing the overall quality of life for individuals diagnosed with ASD. Nevertheless, additional investigation and rigorous clinical trials are imperative to substantiate the effectiveness and safety of these interventions in managing Autism Spectrum Disorder (ASD). In addition it would be beneficial to be referred more sound, that these interventions should applied parallel to other interventions such as Occupational therapy, Speech and Language Therapy and more.

Introduction

It may be beneficial to briefly describe the possible limitations of existing pharmacological interventions and the need for alternative approaches.

Methods

The manuscript should include a section detailing the methodology used (if used) for the review, such as databases and search terms utilized, inclusion/exclusion criteria, and the selection process of the studies included.

Conclusion

The conclusion should succinctly outline the primary findings of the review and underscore the potential efficacy of non-pharmacological interventions in managing Autism Spectrum Disorder (ASD). Additionally, this finding may underscore the necessity for additional research and the significance of individualized and interdisciplinary strategies in ASD intervention.

Additionally, it is imperative to acknowledge and rectify the existing deficiencies in understanding and propose directions for future research in this field.

Author Response

We would like to thanks the reviewer for spending valuable time in reviewing this manuscript and sharing the valuable comments. As per the suggestion manuscript is revised and modifications are highlighted in green color and report is attached. 
